# Privacy Auditing Synthetic Data Release through Local Likelihood Attacks

## Abstract

Auditing the privacy leakage of synthetic data is an important but unresolved problem. Most existing privacy auditing frameworks for synthetic data rely on heuristics and unreasonable assumptions to attack the failure modes of generative models, exhibiting limited capability to describe and detect the privacy exposure of training data through synthetic data release. In this paper, we study designing Membership Inference Attacks (MIAs) that specifically exploit the observation that tabular generative models tend to significantly overfit to certain regions of the training distribution. Here, we propose Generative Likelihood Ratio Attack (Gen-LRA), a novel, computationally efficient No-Box MIA that, with no assumption of model knowledge or access, formulates its attack by evaluating the influence a test observation has in a surrogate model's estimation of a local likelihood ratio over the synthetic data. Assessed over a comprehensive benchmark spanning diverse datasets, model architectures, and attack parameters, we find that Gen-LRA consistently dominates other MIAs for generative models across multiple performance metrics. These results underscore Gen-LRA's effectiveness as a privacy auditing tool for the release of synthetic data, highlighting the significant privacy risks posed by generative model overfitting in real-world applications.[1]

## 1 Introduction

Real world tabular data is often privacy-sensitive to the individual observations that compose these samples, hindering their ability to be shared in open-science efforts that can aid in new research and improve reproducibility. A promise of generative modeling is that models trained on sensitive data can produce samples that preserve the privacy of the training set while maintaining much of its intrinsic statistical information, enabling responsible release to a third party. In practice, a wide array of methodologies have been proposed to accomplish synthetic data release involving modifying loss functions (Abadi et al., 2016; Wang et al., 2022), creating new generative model architectures (Yoon et al., 2019; 2020a), and studying data release strategies (Hardt et al., 2012; Gupta et al., 2012; Takagi et al., 2021) to provide differential privacy guarantee. In another direction, a variety of methods have been proposed that maximize the fidelity of synthetic data and argue that privacy is satisfied through ad-hoc similarity metrics (Zhao et al., 2021; Guillaudeux et al., 2022; Liu et al., 2023; Solatorio and Dupriez, 2023).

To audit the empirical privacy of synthetic data generators, Membership Inference Attacks (MIAs) have recently been extended from traditional machine learning models to synthetic tabular data. Here, privacy auditing is framed as an adversarial game: given specific constraints defined by a threat model, an attacker attempts to determine whether a test observation belongs to a model's training dataset exploiting some notion of model failure (Shokri et al., 2017; Chen et al., 2020; Carlini et al., 2021). A successful attack represents a concrete privacy breach with clear real-world implications, where other similarity-based metrics have been shown to fail to capture privacy risk (Platzer and Reutterer, 2021; Ganev and Cristofaro, 2023; Ward et al., 2024).

While a promising, MIAs for generative models and synthetic data release have seen limited success. Previous work in MIAs for synthetic data release has often relied on distance or density-based heuristics for their attacks or have included additional assumptions about model query access that are

---

[1]An anonymous repository can be found here.

Table 1: Mean (STD) relative rank of each MIA across models, datasets, training sizes, and seeds. As means of MIA metrics can obfuscate their true performance, we report the relative rank of each attack for AUC-ROC and TPR@FPR. We find that if an adversary were to choose Gen-LRA they would ususally have selected the best attack.

| MIA | AUC-ROC | TPR@FPR=0 | TPR@FPR=0.001 | TPR@FPR=0.01 | TPR@FPR=0.1 |
|---|---|---|---|---|---|
| Gen-LRA (ours) | **1.32 (1.04)** | **1.29 (0.83)** | **1.26 (0.81)** | **1.22 (0.77)** | **1.22 (0.77)** |
| DCR | 4.36 (1.94) | 4.25 (0.90) | 4.28 (0.93) | 4.21 (1.11) | 4.17 (1.58) |
| DCR-Diff | 4.29 (1.65) | 4.30 (0.76) | 4.31 (0.79) | 4.32 (0.94) | 4.38 (1.30) |
| DOMIAS | 4.32 (1.73) | 4.44 (0.67) | 4.47 (0.71) | 4.53 (0.83) | 4.48 (1.33) |
| DPI | 4.35 (1.71) | 4.43 (0.72) | 4.39 (0.73) | 4.37 (0.90) | 4.31 (1.31) |
| LOGAN | 4.52 (1.60) | 4.41 (0.75) | 4.44 (0.80) | 4.47 (0.92) | 4.53 (1.30) |
| MC | 4.40 (1.91) | 4.46 (0.85) | 4.44 (0.87) | 4.47 (1.02) | 4.50 (1.55) |

unrealistic to the release setting and computationally do not scale to modern architectures. In contrast, we focus on studying membership inference for the release of synthetic data in a No-Box Threat Model (Houssiau et al., 2022). In this approach, we make no adversarial assumptions of knowledge about model architecture, access, and training parameters that mimics real-world scenarios of parties following best practices for releasing synthetic data in domains like healthcare and finance. Under this threat model, we derive a powerful MIA called Generative Likelihood Ratio Attack (Gen-LRA) which constructs an influence function formulated from likelihood ratio estimation to target privacy leakage that occurs through model overfitting. We show that our attack broadly outperforms competing methods especially at low fixed false positive rates, highlighting that overfitting presents a more dangerous source of privacy leakage than previously suggested. Our contributions are as follows:

**Contributions**:

1. We introduce Gen-LRA, a novel MIA that uses an influence function framework to attack overfitting in tabular generative models with minimal assumptions by evaluating the likelihood ratio of synthetic data under a surrogate model trained with and without a test point.

2. We show that Gen-LRA is computationally efficient and broadly outperforms other MIAs for synthetic data generators across a diverse benchmark of datasets, model architectures, and experiment parameters. (Table 2)

3. We demonstrate that Gen-LRA better identifies subgroups of training observation that experience egregious privacy leakage relative to other attacks (Table 3). We also show that Gen-LRA can be used as an evaluation tool for overfitting in tabular generative models (Figure 2).

## 2 MEMBERSHIP INFERENCE ATTACKS FORMALISM

In this work, we specifically study the Membership Inference Attack Game in the context of *synthetic data release*. The objective of this game is to determine whether a particular data point was included in the original training dataset by examining the outputs of a generative model. We first introduce the formal definition of the *Membership Inference Attack Game*:

**Definition (Membership Inference Attack Game).** The game proceeds between a challenger $\mathcal{C}$ and an adversary $\mathcal{A}$ as follows:

1. The challenger samples a training dataset $T = x_i{}_{i=1}^n$ from the population distribution $x_i \sim \mathbb{P}$ and uses $T$ to train a tabular generative model $\mathcal{G} \leftarrow \mathcal{T}(T)$. The generative model $\mathcal{G}$ produces synthetic dataset $S$.

2. The challenger flips a bit $b \in 0, 1$. If $b = 0$, the challenger samples a test observation $x^\star$ from the population distribution $\mathbb{P}$. Otherwise, the challenger selects the test observation $x^\star$ from the training set $T$.

3. The challenger sends the test observation $x^\star$ to the adversary $\mathcal{A}$.

4. The adversary has access to some information defined by a threat model and uses this information to output a guess $\hat{b} \leftarrow \mathcal{A}(x^\star)$.

5. The output of the game is 1 if $\hat{b} = b$, and 0 otherwise. The adversary wins if $\hat{b} = b$, i.e., if it correctly identifies whether the test observation $x^\star$ was part of the training set $T$ or a freshly sampled data point from the population distribution $\mathbb{P}$.

**Adversary's Goal and Capabilities**   The adversary $\mathcal{A}$ in the Membership Inference Game aims to determine whether a specific data point $x^\star$ was part of the original training dataset $T$ or was drawn from the population distribution $\mathbb{P}$. Here, the adversary can utilize available information in any manner to construct a method to classify the membership of $x^*$. The performance of the classifier, which can be evaluated with binary classification metrics, is a measure of the privacy leakage of the training data from $\mathcal{G}$ through $S$. Formally, this classification or Membership Inference Attack can be expressed as:

$$\mathcal{A}(x^\star) = \mathbb{I}\left[f(x^\star) > \gamma\right] \tag{1}$$

where $\mathbb{I}$ is the indicator function, $f(x^\star)$ is a scoring function of $x^*$, and $\gamma$ is an adjustable decision threshold.

**Threat Model**   In this paper, we consider a "No-box" (Houssiau et al., 2022) threat model where the adversary is assumed to have no access to the internal structure, parameters, or sampling mechanism of the generative model. Instead, the attack must be constructed using only two observed datasets: the released synthetic dataset $S \sim \mathcal{G}(T)$, and an independently collected reference dataset $R \sim \mathbb{P}$ drawn from the same underlying population. The auditor is not granted access to the training set $T$, nor to labeled membership indicators, and cannot issue queries to the generator. This reflects deployment scenarios in which organizations release synthetic data for downstream analysis while keeping all model knowledge confidential. The synthetic dataset $S$ serves as the only potential leakage surface, and the reference set $R$ provides a statistical anchor for the population. This reference dataset is often assumed in No-box attacks for synthetic data Chen et al. (2020); Houssiau et al. (2022); van Breugel et al. (2023); Ward et al. (2024) as well as generally for supervised learning models (Carlini et al., 2021; Ye et al., 2022; Zarifzadeh et al., 2024) and represents a kind of 'worst case' scenario where an adversary may be able to find comparable data in the real world such as open source datasets, paid collection, prior knowledge, etc.

**Attack Strategy**   The adversary must develop a strategy in which to construct Equation (1). We specifically propose that the adversary utilize the *degree of local overfitting* within $S$ as the primary signal to determine whether a specific data point $x^\star$ belongs to the training set.

Overfitting is a common and difficult-to-eliminate failure mode in generative models, particularly in the context of tabular synthetic data generation. In the setting of Membership Inference Attacks, this failure mode becomes a significant source of privacy leakage. van Breugel et al. (2023) for example identified that TVAE (Xu et al., 2019) overfit to minority class examples in a medical training dataset, leaking their privacy. Similarly, Ward et al. (2024) found that TabDDPM (Kotelnikov et al., 2022), when tasked with generating synthetic data for the well-known Adult dataset, heavily replicated data points from certain demographic groups within the training data. The key insight drawn from this phenomenon is that areas of the synthetic data distribution with higher density are likely to reflect signals from the original training data. Leveraging this failure, it becomes possible to infer whether specific data points were part of the training set, thus providing a basis for designing privacy attacks. Our work builds on these findings by proposing a new method to measure the degree of local overfitting in generative models. We utilize this metric to design a Membership Inference Attack aimed at exposing the potential privacy risks inherent in synthetic data (See Section 3).

## 3   GENERATIVE LIKELIHOOD RATIO ATTACK

In this section, we propose Generative Likelihood Ratio Attack (Gen-LRA), a powerful MIA designed to detect membership leakage in synthetic data through a statistical notion of *likelihood influence*. Unlike usual MIAs that evaluate the density or distance of a test point itself, Gen-LRA poses membership inference as a function designed to evaluate the influence of $x^\star$ on an estimate of the likelihood of $S$. Here, the central idea is that if $x^\star$ was in the training data and the generative model

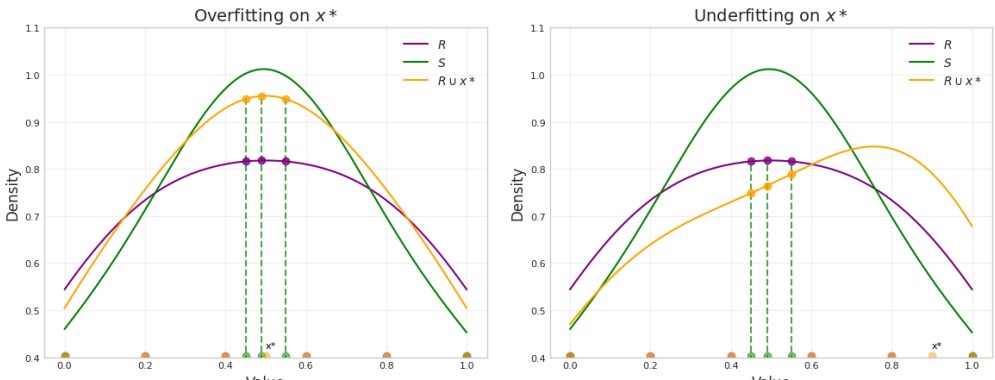

Figure 1: A geometric intuition for Gen-LRA with a 1-dimensional toy example. We visualize the KDE plots of $R, R \cup x^*, S$ as well as the estimated densities of the synthetic observations over $R$ and $R \cup x^*$. Left: we consider $x^* = 0.5$. In this example, the likelihood of the synthetic observations (product of orange intersections) are higher under the density estimate of $R \cup x^*$ than $R$ (product of purple intersections) and therefore we conclude that $x^* \in T$. Right: where $x^* = 0.9$, the opposite is true and we therefore conclude $x^* \notin T$.

is overfit, its inclusion to a surrogate density estimator should increase the estimated likelihood of the synthetic dataset.

## 3.1 EMPIRICAL INFLUENCE FUNCTIONS

Before formalizing our approach, we introduce the concept of influence functions, which provides the theoretical foundation for our attack. Originally developed in robust statistics (Hampel, 1974; Cook and Weisberg, 1986), influence functions measure how statistical estimates change when the underlying data distribution is perturbed. The influence function for an estimator $\theta$ applied to a distribution $F$ is defined as: $\mathcal{I}(x^*, F, \theta) = \lim_{\epsilon \to 0} \frac{\theta((1-\epsilon)F + \epsilon\delta_{x^*}) - \theta(F)}{\epsilon}$ where $\delta_{x^*}$ is the Dirac measure placing mass 1 at point $x^*$.

This definition captures the sensitivity of $\theta$ to infinitesimal perturbations in $F$ at the point $x^*$. Intuitively, it measures how the estimator would change if we slightly increased the probability of observing $x^*$. In the empirical setting with finite samples, influence can be measured by evaluating how estimates change when adding or removing a specific point. Let $\mathcal{D} = \{x_1, x_2, ..., x_n\}$ be a dataset of $n$ samples. The empirical influence function is defined as:

$$\hat{\mathcal{I}}(x^*, \mathcal{D}, \theta) = \theta(\mathcal{D} \cup \{x^*\}) - \theta(\mathcal{D}) \tag{2}$$

For supervised learning models, influence is typically measured as a difference in loss or empirical risk of models trained with and without $x^*$ Koh and Liang (2017). In an MIA for tabular generative models that assumes no model access however, measures of loss are not readily available. Rather than examining how $x^*$ affects model parameters directly, Gen-LRA instead considers the influence on the likelihood assigned to generated samples $S$ over a surrogate estimator.

## 3.2 LIKELIHOOD INFLUENCE AS AN ATTACK SURFACE

To begin, recall that $T, R \sim \mathbb{P}$, our goal is to infer if $x^* \in T$ given $S$ and $R$ based on Equation 1, and that we hypothesize that due to model failure, $S$ is overfit to $x^*$. Gen-LRA measures this overfitness to $x^*$ by formalizing an influence function defined as the difference in the estimated likelihood ratio of the synthetic dataset under two surrogate models: one trained on the original reference dataset $R$, and another on an augmented dataset $R \cup \{x^\star\}$. We define this influence function as:

$$\hat{\mathcal{I}}(x^\star; R, S) := \log \hat{p}(S \mid R \cup \{x^\star\}) - \log \hat{p}(S \mid R), \tag{3}$$

where $\hat{p}$ is the estimated probability. Intuitively (see Figure 1), if the inclusion of $x^\star$ leads to a significant increase in the likelihood of $S$ under a surrogate model, it suggests that $x^\star$ likely

contributed to the generative process. If the likelihood is unchanged or decreases, it implies $x^* \notin T$. In principle, this influence function does not necessarily need to be a measure of the likelihood ratio. However, there are several advantages relative to other options in that this formulation allows for a great amount of flexibility in tuning the attack, and that the likelihood ratio is invariant to encodings of the data.

**Theorem 3.1.** *Let $S$, $R$ be sets of samples and $x^\star$ a new sample point with probability distributions on $\mathcal{X}$. Define:*

$$\hat{\mathcal{I}}(x^\star; R, S) = \log p(S \mid R \cup \{x^\star\}) - \log p(S \mid R) \tag{4}$$

*For any invertible function $g : \mathcal{X} \to \mathcal{X}$, the log-likelihood ratio is invariant:*

$$\hat{\mathcal{I}}(g(x^\star), g(R), g(S)) = \hat{\mathcal{I}}(x^\star, R, S) \tag{5}$$

We refer to Appendix 1.1 for the proof.

### 3.3 GEN-LRA IMPLEMENTATION

Having established our influence function in Equation 3, we can directly utilize this measurement as the scoring function in our membership inference framework from Equation 1, such that $f(x^\star) = \hat{\mathcal{I}}(x^\star; R, S)$. However, the practical deployment of Gen-LRA requires calibration of how we estimate $\hat{\mathcal{I}}(x^\star; R, S)$ to optimize attack performance. Below, we detail the key implementation strategies that enable us to achieve maximum discriminative power when distinguishing between training and non-training samples. A corresponding description of the full algorithm can be found in Appendix 2.1.

**Localization**    A common theme in designing MIAs is to adopt techniques that maximize the signal of $x^*$'s membership in the attack. Realistically, there is likely to be very little signal in comparing the likelihoods of $S$ over estimated probability density functions with a difference of a single observation. Indeed, (3) is an attack over the global likelihood of $S$ which may not be sensitive to detecting subtle patterns of *local* overfitting. Here, we *localize* Gen-LRA by only considering the $k$-nearest elements in $S$ to $x^*$ in our estimation. In practice, the choice of $k$ can have minor impacts on the effectiveness of the attack, but we find we get excellent results with low values of $k$ (See Appendix 4.1).

**Choice of Surrogate Model**    In principle, most density estimation techniques such as tractable probabilistic models (De Cao et al., 2019; Kobyzev et al., 2021; Liu and Van den Broeck, 2021) and Bayesian methods (Hjort, 1996; Grazian and Fan, 2020) can be used to estimate Equation 3. We find though that many of these methods are unsuccessful at estimating this likelihood ratio given a one unit observation difference. As a rule of thumb, we use Gaussian Kernel Density Estimators (KDEs) (Węglarczyk, Stanisław, 2018) as they are widely known, computationally cheap, and achieve state of the art results. We also find in our experimentation that KDEs empirically outperform De Cao et al. (2019), a leading deep-learning-based density estimator (see Section 6.3).

**Choice of Decision Threshold**    While Section 3.2 details the derivation of a scoring function $f(x^*)$, (1) still requires a decision threshold $\gamma$. Intuitively for Gen-LRA, the decision threshold $\gamma$ can be any chosen threshold but $\hat{\mathcal{I}}(S, R, x^\star) > 1$ implies some degree of local overfitting to $x^*$.

## 4 RELATED WORKS

### 4.1 ASSESSING OVERFITTING IN TABULAR GENERATIVE MODELS

Several measures have been developed to assess the fitness of tabular synthetic data, particularly from a privacy perspective. These metrics generally aim to measure the similarity between the training and synthetic datasets, with the ideal outcome being that the synthetic data is neither too similar to the training data nor too different. A widely used metric for this purpose is Distance to Closest Record[2] (Park et al., 2018; Lu et al., 2019; Yale et al., 2019; Zhao et al., 2021; Guillaudeux

---

[2]DCR in the similarity metric case compares a training point to a synthetic point. However, Chen et al. (2020) proposes an MIA where the scoring function is a distance computation for a test point and a synthetic point. In all other sections of the paper we use DCR to refer to the MIA.

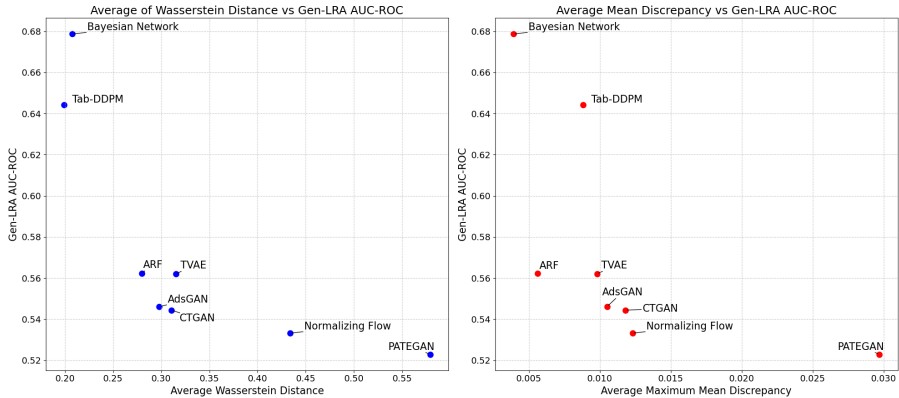

Figure 2: Average Wasserstein Distance and Average Maximum Mean Discrepancy plotted against Gen-LRA AUC-ROC for benchmarked models. Bayesian Network and Tab-DDPM outperform other models in these performance metrics but have higher privacy risk highlighting that Gen-LRA can be used to characterize a privacy-utility tradeoff in tabular generative models.

et al., 2022; Liu et al., 2023), which compares the distance from each training point to its nearest neighbor in the synthetic dataset to which a mean is computed. Another commonly used metric is the Identical Matching Score (IMS) (Lu et al., 2019; AI, 2020; 2021), which measures the proportion of identical records between the training and synthetic datasets. While these measures can be useful for describing overfitness from a distribution-level quality and model generalization perspective, they do not characterize privacy risk because there is no assumed threat model and they are not evaluated over non-member examples.

## 4.2 MIAs for Tabular Generative Models

Membership Inference Attacks on the other hand, explicitly characterize the empirical privacy risk of a machine learning model (Yeom et al., 2018; Song and Mittal, 2020). Originally, MIAs were developed for attacking supervised learning classifiers (Shokri et al., 2017). In this context, the general idea for these attacks is to query a model with different observations to learn patterns in its class probability outputs. Membership can then be inferred by comparing the outputs of the model to outputs from reference models in some manner (Sablayrolles et al., 2019; Long et al., 2020; Carlini et al., 2021; Watson et al., 2022; Ye et al., 2022; Zarifzadeh et al., 2024).

To adapt to these structural differences, a wide range of MIAs for tabular generative models have been proposed that utilize different threat models and strategies to construct (1) (Hayes et al., 2017; Hilprecht et al., 2019; Chen et al., 2020; Stadler et al., 2022; Houssiau et al., 2022; van Breugel et al., 2023; Meeus et al., 2024; Ward et al., 2024). Of these, Gen-LRA is most related to DOMIAS van Breugel et al. (2023) and a line of work that extends query-based attacks to tabular generative models Stadler et al. (2022); Houssiau et al. (2022); Meeus et al. (2024).

DOMIAS follows the same threat model assumptions and has a similar construction to Gen-LRA defining its scoring function in (1) as a density ratio $\frac{p_S(x^*)}{p_R(x^*)}$. Gen-LRA however improves on DOMIAS in that the score for DOMIAS can only be a single point estimate whereas Gen-LRA can be comprised of many estimates of a local region, allowing it to incorporate more information. Furthermore, Gen-LRA measures the effect of the specific inclusion of $x^*$ on $S$, which is more proximal to the membership inference problem than measuring the density of $x^*$ from $S$. These differences allow Gen-LRA to broadly outperform DOMIAS in our experimentation.

In another direction, Stadler et al. (2022); Houssiau et al. (2022); Meeus et al. (2024) propose query-based attacks on tabular generators where they additionally assume an adversary has knowledge of the *implementation* of target model. In these methods, an attacker trains many versions of the model with $R \cup x^*$ and $R$ which are used to generate many synthetic datasets. Summary statistics and histograms are then constructed to represent each synthetic dataset as a vector and a classifier is then

Table 2: Mean (STD) AUC-ROC for each Membership Inference Attack across model architectures and datasets. Gen-LRA outperforms all other threat-model comparable attacks with an average rank of 1 across all architectures.

| Model | Gen-LRA (Ours) | DCR-Diff | DOMIAS | DPI | DCR | MC | LOGAN |
|---|---|---|---|---|---|---|---|
| AdsGAN | **0.534 (0.02)** | 0.517 (0.02) | 0.517 (0.02) | 0.521 (0.02) | 0.516 (0.02) | 0.515 (0.02) | 0.503 (0.02) |
| ARF | **0.562 (0.03)** | 0.540 (0.02) | 0.534 (0.02) | 0.538 (0.02) | 0.533 (0.02) | 0.527 (0.02) | 0.504 (0.02) |
| Bayesian Network | **0.679 (0.07)** | 0.656 (0.06) | 0.632 (0.06) | 0.557 (0.02) | 0.665 (0.07) | 0.625 (0.05) | 0.505 (0.02) |
| CTGAN | **0.533 (0.02)** | 0.515 (0.02) | 0.515 (0.02) | 0.519 (0.02) | 0.513 (0.02) | 0.511 (0.02) | 0.504 (0.02) |
| Normalizing Flows | **0.524 (0.02)** | 0.504 (0.02) | 0.505 (0.02) | 0.506 (0.02) | 0.505 (0.02) | 0.504 (0.02) | 0.502 (0.02) |
| PATEGAN | **0.520 (0.02)** | 0.497 (0.02) | 0.498 (0.02) | 0.500 (0.02) | 0.500 (0.02) | 0.501 (0.02) | 0.502 (0.02) |
| Tab-DDPM | **0.603 (0.08)** | 0.587 (0.06) | 0.587 (0.06) | 0.552 (0.03) | 0.585 (0.07) | 0.564 (0.05) | 0.505 (0.02) |
| TabSyn | **0.583 (0.04)** | 0.553 (0.02) | 0.561 (0.06) | 0.547 (0.06) | 0.585 (0.07) | 0.517 (0.05) | 0.501 (0.02) |
| TVAE | **0.541 (0.02)** | 0.529 (0.03) | 0.524 (0.03) | 0.523 (0.02) | 0.529 (0.03) | 0.522 (0.02) | 0.504 (0.02) |
| **Average Rank** | **1.0** | 3.4 | 3.6 | 3.8 | 3.8 | 5.34 | 6.4 |

trained using these representations to differentiate between synthetic datasets trained from $R \cup x^*$ and $R$ respectively.

These attacks are related to Gen-LRA as they all aim to estimate the likelihood ratio of (3), but Gen-LRA improves upon them in two main ways. First, these attacks are unsuitable for auditing privacy in synthetic data release as they are trivially easy to defeat because the defender can choose to just not release the implementation of the architecture they used to generate the synthetic data. Indeed, Golob et al. (2024) has shown that there can be significant privacy leakage in differentially private synthetic data generation from this exact scenario such that best practice for data releasing parties is to disclose as little model information as possible. Gen-LRA makes no assumption about model implementation and thus follows a more realistic threat model for synthetic data release. Secondly, these attacks are computationally expensive as they rely on training many surrogate models for each $x^*$ to construct their attack. In practice, it is impractical to train $(N_{TestSetSampleSize} + 1) * N_{SurrogateModels}$ separate models to audit a single trained model, especially as large diffusion and language model architectures become more popular. Gen-LRA instead only requires a total of $N_{TestSetSampleSize} + 1$ density estimators to be fit which is much cheaper.

## 5 EXPERIMENTS

### 5.1 BENCHMARKING

We evaluate Gen-LRA's effectiveness across a benchmark of 15 tabular datasets, 7 membership inference attacks (MIAs), and 9 tabular generative models (full details on MIAs, architectures, and datasets are in Appendix 3.1). For each dataset, we randomly sample without replacement three equal-sized sets: training $T$, reference $R$, and holdout $H$. The training set is used to train each architecture, which then generates an equally sized synthetic dataset. MIAs are evaluated using one-hot and scaled encodings from the synthetic data to prevent data leakage. We repeat this process over 10 seeds for each dataset with sample sizes of N = (250, 1000, 4000).

Since DOMIAS and Gen-LRA rely on density estimation techniques, we implement these methods using Gaussian Kernel Density Estimation (KDE), as we find KDE with a Silverman's Rule bandwidth parameter outperforms deep learning-based estimators (see Section 3.3). Since KDE can struggle with one-hot encoded categorical data, we use ordinal encoding for these MIAs. We present an ablation study in Appendix 4.1 with various PCA and VAE-based encoding strategies, though our experiments show ordinal encoding sees the best performance. For Gen-LRA, we found that the locality parameter $k$ has a modest impact on attack performance (see Appendix 4.2), so we set $k = 10$ throughout our experiments.

**Baselines** We compare Gen-LRA against all MIAs for Tabular Synthetic data that follow compatible threat models: LOGAN, MC, DCR/DCR Difference, DOMIAS, and DPI (Hayes et al., 2017; Hilprecht et al., 2019; Chen et al., 2020; van Breugel et al., 2023; Ward et al., 2024). For synthetic data architectures, we evaluate across nine models: Bayesian Network (BN), PATEGAN, Ads-GAN, CTGAN, TVAE, Normalizing Flows (NFlows), ARF, Tab-DDPM, and TabSyn (Ankan and Panda, 2015; Yoon et al., 2019; 2020b; Xu et al., 2019; Durkan et al., 2019; Watson et al., 2023;

Table 3: Mean (STD) True Positive Rates for MIAs at different fixed False Positive Rate levels across experiment runs. Gen-LRA outperforms other threat-model compatible MIAs.

| MIA | TPR@FPR = 0.001 | TPR@FPR = 0.01 | TPR@FPR = 0.1 |
|---|---|---|---|
| LOGAN | 0.003 (0.01) | 0.012 (0.01) | 0.102 (0.02) |
| DPI | 0.002 (0.00) | 0.014 (0.01) | 0.118 (0.03) |
| MC | 0.003 (0.00) | 0.014 (0.01) | 0.120 (0.04) |
| DOMIAS | 0.002 (0.00) | 0.016 (0.01) | 0.134 (0.06) |
| DCR-Diff | 0.005 (0.01) | 0.019 (0.02) | 0.138 (0.07) |
| DCR | 0.016 (0.05) | 0.036 (0.08) | 0.153 (0.11) |
| Gen-LRA (ours) | **0.031 (0.01)** | **0.056 (0.03)** | **0.193 (0.08)** |

Kotelnikov et al., 2022; Zhang et al., 2024).) For TabSyn, we use the original implementation with default hyperparameters, for all other architectures we use the default Synthcity (Qian et al., 2023) implementations.

All experiments were conducted on an AWS G5.2xlarge EC2 instance. The main experimental findings took approximately 72 hours of compute on this system between data generation and auditing. Additional compute was used in preliminary and secondary experiments, especially those described in Section 6.3 which was approximately 80 hours of compute.

# 6 DISCUSSION

## 6.1 GEN-LRA PERFORMANCE

Gen-LRA is a density-based attack that, using a simple estimation strategy, broadly outperforms competing methods (Tables 1, 2, 3). Constructing the attack as a likelihood ratio over local regions of the synthetic probability distribution allows greater attack performance as Gen-LRA is customizable in its choice of $k$ to different datasets and architectures. Indeed as Table 2 shows, models like Tab-DDPM and Bayesian Networks experience more privacy leakage than others and a tunable attack can realize large performance gains. While Gen-LRA excels in a global attack evaluation setting demonstrating that on average it outperforms all other attacks across all model architectures with an average rank of 1. We additionally compare the average AUC-ROC for each architecture from Gen-LRA to measures of model performance in Figure 2. We find that models with higher performance also exhibit greater privacy leakage. This showcases that Gen-LRA can be used in model benchmarking to characterize a privacy-performance tradeoff for synthetic data generation. Lastly, we evaluate the relative rank for each MIA across experiment runs in Table 1 and find that Gen-LRA dominates other attacks with an average relative rank of 1.32 for AUC-ROC.

## 6.2 THE LOW FALSE POSITIVE SETTING

While AUC-ROC provides an easily comparable, well-understood global measure of an attack's effectiveness, from a privacy perspective it does not indicate how well an attack performs when the False Positive Rate (FPR) is low. As Carlini et al. (2021) and Zarifzadeh et al. (2024) argue, researchers should analyze how well an attack performs with a low FPR because in practical settings there is a greater privacy risk to individual training observations that can be correctly classified with few false positives versus observations that are included with many false positives.

We therefore report the mean and standard deviation TPR@FPRs (True Positive Rate at False Positive Rate) for a range of fixed FPR values for each MIA across datasets, architectures, and $N$-sizes available in Table 3. Achieving a high TPR at a very low FPR is challenging in this scenario, however, Gen-LRA nearly doubles the performance of the next best method at FPR = 0.001 and consistently sees significant gains over the next best method at higher thresholds. This highlights that Gen-LRA is better able to detect egregious overfitting to certain training observations, relative to other competing attacks at comparable threat models.

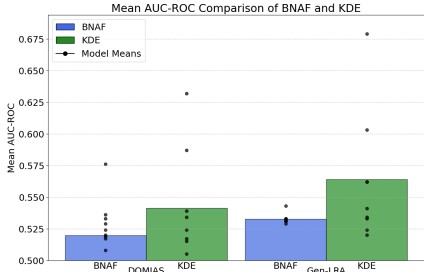

Figure 3: A comparison of the Mean AUC-ROC for DOMIAS and Gen-LRA using density estimation techniques BNAF and KDE. The group mean performance for each model are also plotted on each attack/ estimation bar. Overall, we see that KDE outperforms BNAF for both DOMIAS and Gen-LRA. While the variance of performance across models is less with BNAF than KDE for both attacks, KDE outperforms BNAF on models that exhibit more egregious privacy leakage (Bayesian Network and Tab-DDPM) whereas BNAF fails to identify it.

### 6.3 DEEP LEARNING DENSITY ESTIMATION

As Gen-LRA relies on estimating the likelihood of high dimensional data, it is surprising that it excels with using Gaussian Kernel Density Estimation (KDE), which is a baseline that is usually outperformed by more modern density estimation methods in metrics such Average Negative Log Likelihood and Negative Evidence Lower Bound (De Cao et al., 2019; Wen and Hang, 2022). We repeat this benchmarking experiment, and following van Breugel et al. (2023), we use Block Neural Autoregressive Flows (BNAF) to study the performance of DOMIAS and Gen-LRA with a more powerful deep-learning-based density estimation technique.

We visualize these results in Figure 3 where we find that KDE actually better identifies privacy leakage than BNAF for DOMIAS and Gen-LRA. Both of these attacks rely on estimating subtle differences in the densities of local regions for two separately learned but similar probability distributions. We hypothesize KDE could be better suited for the task of privacy auditing because it fits locally based on its bandwidth parameter, whereas BNAF learns the global distribution using many sensitive hyperparameters that can effect its performance. In any case, in all other experiments we default to reporting the KDE version of DOMIAS and Gen-LRA and we recommend practitioners use KDE for these methods as empirically it is better at identifying extreme cases of privacy leakage and is also substantially less computationally expensive to run versus BNAF.

## 7 CONCLUSION

Membership Inference Attacks are a useful tool for privacy auditing generative models for synthetic data release. They can characterize the privacy risk towards training observations, provide information on how a model may be overfit, and add subtle context to patterns of behavior in generative models. In this paper, we propose Gen-LRA, which attacks synthetic data by a evaluating a likelihood ratio designed to detect overfitting. We show that Gen-LRA excels at attacking a diverse set of generative models across a wide-range of datasets and that this success comes from Gen-LRA's ability to target a generative model's tendency to overfit to training data relative to a broader population distribution. We note that a limiation with Gen-LRA in that it requires hyperparameters based on its localization and density estimation strategies. However, we point out that empirically, Gen-LRA usually outperforms other attacks despite these disadvantages and is widely compatible with many application or domain-specific density estimation techniques.

We believe that there are many directions for future work. Exploring emerging density estimation methodologies would likely yield better empirical performance, especially on high dimensional datasets. On a different front, research into developing adversarial techniques to better understand model overfitting in general could also lead to important interpretability techniques. Lastly, we believe that while tabular data generators provide a solution to the common privacy problem of data sharing, more work needs to be done to develop practical auditing methodologies practitioners can follow to audit potential security vulnerabilities in this emerging technology.

## 8 STATEMENT OF ETHICS

The ability of adversaries to infer whether an individual's data was part of the original dataset poses risks to privacy, particularly in domains like healthcare, finance, and social sciences, where sensitive personal data is frequently used. If synthetic data does not sufficiently obfuscate membership information, it could lead to re-identification risks. While this work proposes one such re-identification method, its ultimate goal is to help researchers and practitioners to conduct more powerful privacy assessments before deploying synthetic datasets. We believe adversarial work is critical for the research and development of better privacy systems.

## 9 STATEMENT OF REPRODUCIBILITY

We make our code available at this link which facilitates running our main experiments. Furthermore we provide dataset, generator, comparison MIA and a full Gen-LRA algorithm descriptions in the Appendix.

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

## APPENDIX

## A   PROOFS

**Theorem A.1.** *Let $S$ be a set of samples, $R$ a reference set, and $x^\star$ a new sample point, with probability distributions defined on $\mathcal{X}$. Define the log-likelihood ratio:*

$$\Delta(S, R, x^\star) = \log p(S \mid R \cup \{x^\star\}) - \log p(S \mid R) \tag{6}$$

*Let $g : \mathcal{X} \to \mathcal{X}, x \mapsto g(x)$ be some invertible function, and define transformed sets $\tilde{S} = g(S)$, $\tilde{R} = g(R)$, and $\tilde{x}^\star = g(x^\star)$ with respective distributions $\tilde{p}$. Then $\Delta(\tilde{S}, \tilde{R}, \tilde{x}^\star) = \Delta(S, R, x^\star)$, i.e., the same log-likelihood ratio is obtained for either data representation.*

*Proof.* Similarly to van Breugel et al. (2023), using the change of variables formula, we have $\tilde{p}(g(A)) = \frac{p(A)}{|J(A)|}$ with Jacobian $J(x) = \frac{dg}{dx}(x)$ for any set $A$.

For the conditional probabilities, we have:

$$\tilde{p}(\tilde{S} \mid \tilde{R} \cup \{\tilde{x}^\star\}) = \frac{\tilde{p}(\tilde{S}, \tilde{R} \cup \{\tilde{x}^\star\})}{\tilde{p}(\tilde{R} \cup \{\tilde{x}^\star\})} \tag{7}$$

$$= \frac{p(S, R \cup \{x^\star\})/|J(S, R \cup \{x^\star\})|}{p(R \cup \{x^\star\})/|J(R \cup \{x^\star\})|} \tag{8}$$

Similarly:

$$\tilde{p}(\tilde{S} \mid \tilde{R}) = \frac{\tilde{p}(\tilde{S}, \tilde{R})}{\tilde{p}(\tilde{R})} \tag{9}$$

$$= \frac{p(S, R)/|J(S, R)|}{p(R)/|J(R)|} \tag{10}$$

Since $J(S, R \cup \{x^\star\}) = J(S, R)$ and $J(R \cup \{x^\star\}) = J(R)$ (the Jacobians are the same when operating on spaces of the same dimension), we have:

$$\frac{\tilde{p}(\tilde{S} \mid \tilde{R} \cup \{\tilde{x}^\star\})}{\tilde{p}(\tilde{S} \mid \tilde{R})} = \frac{p(S, R \cup \{x^\star\})/|J(S, R)|}{p(R \cup \{x^\star\})/|J(R)|} \cdot \frac{p(R)/|J(R)|}{p(S, R)/|J(S, R)|} \tag{11}$$

$$= \frac{p(S, R \cup \{x^\star\})}{p(R \cup \{x^\star\})} \cdot \frac{p(R)}{p(S, R)} \tag{12}$$

$$= \frac{p(S \mid R \cup \{x^\star\})}{p(S \mid R)} \tag{13}$$

Taking logarithms:

$$\log \frac{\tilde{p}(\tilde{S} \mid \tilde{R} \cup \{\tilde{x}^\star\})}{\tilde{p}(\tilde{S} \mid \tilde{R})} = \log \frac{p(S \mid R \cup \{x^\star\})}{p(S \mid R)} \tag{14}$$

$$\tag{15}$$

Which gives us:

$$\Delta(\tilde{S}, \tilde{R}, \tilde{x}^\star) = \Delta(S, R, x^\star) \tag{16}$$

as desired. $\square$

## B  ALGORITHM

---

**Algorithm 1** Gen-LRA

---

1: **function** GEN-LRA($X_{\text{test}}, S, R, k$)
2:     $A_{\text{scores}} \leftarrow \emptyset$                                                    ▷ Initialize score array
3:     $\text{DE}_R \leftarrow \text{FitDensityEstimator}(R)$
4:     **for** $x \in X_{\text{test}}$ **do**
5:         $R' \leftarrow R \cup \{x\}$
6:         $\text{DE}_{R'} \leftarrow \text{FitDensityEstimator}(R')$
7:         $S_{\text{close}} \leftarrow \text{FindKNearestNeighbors}(S, x, k)$
8:         $L_{R'} \leftarrow \text{DE}_{R'}(S_{\text{close}})$
9:         $L_R \leftarrow \text{DE}_R(S_{\text{close}})$
10:         $a \leftarrow \sum_{\mathbf{s} \in \mathbf{S}_{\text{close}}} \log(\mathbf{L_{R'}}[\mathbf{s}]) -$
11:             $\sum_{\mathbf{s} \in \mathbf{S}_{\text{close}}} \log(\mathbf{L_R}[\mathbf{s}])$
12:         $A_{\text{scores}} \leftarrow A_{\text{scores}} \cup \{a\}$
13:     **end for**
14:     **return** $A_{\text{scores}}$
15: **end function**

---

## C  EXPERIMENTS/ REPLICATION DETAILS

### C.1  MIAs FOR GENERATIVE MODELS DESCRIPTIONS

The Membership Inference Attacks referenced in this paper is are described as follows:

- **LOGAN** Hayes et al. (2017): LOGAN consists of black box and shadow box attack. The black-box version involves training a Generative Adversarial Network (GAN) on the synthetic dataset and using the discriminator to score test data. A calibrated version improves upon this by training a binary classifier to distinguish between the synthetic and reference dataset. In this paper, we only benchmark the calibrated version.

- **Distance to Closest Record (DCR) / DCR Difference** Chen et al. (2020): DCR is a black-box attack that scores test data based on a sigmoid score of the distance to the nearest neighbor in the synthetic dataset. DCR Difference enhances this approach by incorporating a reference set, subtracting the distance to the closest record in the reference set from the synthetic set distance.

- **MC** Hilprecht et al. (2019): MC is based on counting the number of observations in the synthetic dataset that fall into the neighborhood of a test point (Monte Carlo Integration). However, this method does not consider a reference dataset, and the choice of distance metric for defining a neighborhood is a non-trivial hyperparameter to tune.

- **DOMIAS** van Breugel et al. (2023): DOMIAS is a calibrated attack which scores test data by performing density estimation on both the synthetic and reference datasets. It then calculates the density ratio of the test data between the learned synthetic and reference probability densities.

- **DPI** Ward et al. (2024): DPI computes the ratio of $k$-Nearest Neighbors of $x^*$ in the synthetic and reference datasets. It then builds a scoring function by computing the ratio of the sum of data points from each class of neighbors from the respective sets.

### C.2  GENERATIVE MODEL ARCHITECTURE DESCRIPTIONS

In all experiments, we use the implementations of these models from the Python package Synthcity Qian et al. (2023). For benchmarking purposes, we use the default hyperparameters for each model. A brief description of each model is as follows:

- **CTGAN** Xu et al. (2019): Conditional Tabular Generative Adversarial Network uses a GAN framework with conditional generator and discriminator to capture multi-modal distributions. It employs mode normalization to better learn mixed-type distributions.

- **TVAE** Xu et al. (2019): Tabular Variational Auto-Encoder is similar to CTGAN in its use of mode normalizing techniques, but instead of a GAN architecture, it employs a Variational Autoencoder.

- **Normalizing Flows (NFlows)** Durkan et al. (2019): Normalizing flows transform a simple base distribution (e.g., Gaussian) into a more complex one matching the data by applying a sequence of invertible, differentiable mappings.

- **Bayesian Network (BN)** Ankan and Panda (2015): Bayesian Networks use a Directed Acyclic Graph to represent the joint probability distribution over variables as a product of marginal and conditional distributions. It then samples the empirical distributions estimated from the training dataset.

- **Adversarial Random Forests (ARF)** Watson et al. (2023): ARFs extend the random forest model by adding an adversarial stage. Random forests generate synthetic samples which are scored against the real data by a discriminator network. This score is used to re-train the forests iteratively.

- **Tab-DDPM** Kotelnikov et al. (2022): Tabular Denoising Diffusion Probabilistic Model adapts the DDPM framework for image synthesis. It iteratively refines random noise into synthetic data by learning the data distribution through gradients of a classifier on partially corrupted samples with Gaussian noise.

- **PATEGAN** Yoon et al. (2019): The PATEGAN model uses a neural encoder to map discrete tabular data into a continuous latent representation which is sampled from during generation by the GAN discriminator and generator pair.

- **Ads-GAN** Yoon et al. (2020b): Ads-GAN uses a GAN architecture for tabular synthesis but also adds an identifiability metric to increase its ability to not mimic training data.

- **TabSyn** Zhang et al. (2024)

## C.3 BENCHMARKING DATASETS REFERENCES

We provide the URL for the sources of each dataset considered in the paper. We use datasets common in the tabular generative modeling literature Suh et al. (2023) TabSyn uses a Variational Auto-Encoder to learn a latent space in which it builds a diffusion model from. TabSyn usually achieves state of the art data quality metrics relative to other methods compared.

1. **Abalone** (OpenML): `https://www.openml.org/search?type=data&sort=r uns&id=183&status=active`

2. **Adult** Becker and Kohavi (1996)

3. **Bean** (UCI): `https://archive.ics.uci.edu/dataset/602/dry+bean+d ataset`

4. **Churn-Modeling** (Kaggle): `https://www.kaggle.com/datasets/shrutime chlearn/churn-modelling`

5. **Faults** (UCI): `https://archive.ics.uci.edu/dataset/198/steel+plat es+faults`

6. **HTRU** (UCI): `https://archive.ics.uci.edu/dataset/372/htru2`

7. **Indian Liver Patient** (Kaggle): `https://www.kaggle.com/datasets/uciml/ indian-liver-patient-records?resource=download`

8. **Insurance** (Kaggle): `https://www.kaggle.com/datasets/mirichoi0218/i nsurance`

9. **Magic** (Kaggle): `https://www.kaggle.com/datasets/abhinand05/magic -gamma-telescope-dataset?resource=download`

10. **News** (UCI): `https://archive.ics.uci.edu/dataset/332/online+new s+popularity`

11. **Nursery** (Kaggle): `https://www.kaggle.com/datasets/heitornunes/nu rsery`

12. **Obesity** (Kaggle): `https://www.kaggle.com/datasets/tathagatbanerj ee/obesity-dataset-uci-ml`

13. **Shoppers** (Kaggle): `https://www.kaggle.com/datasets/henrysue/onlin e-shoppers-intention`

14. **Titanic** (Kaggle): `https://www.kaggle.com/c/titanic/data`

15. **Wilt** (OpenML): `https://www.openml.org/search?type=data&sort=run s&id=40983&status=active`

# D  ADDITIONAL RESULTS

## D.1  GEN-LRA ENCODING

As our main experiment uses Kernel Density Estimation (KDE) over (usually) heterogeneous datasets, we present an ablation for encoding tabular data to be numeric such that KDE can converge. We experiment with 3 common strategies used in the density estimation literature: ordinal encoding for categorical variables, one-hot encoding categorical variables and then performing Principle Component Analysis (PCA), and using a Variational Auto-Encoder to learn continuous latent representations of the data.

We repeat our main experiment on TabSyn with these three encoding schemes. For PCA we use the number of eigenvectors that explain up to 95 %variance and for the VAE encoding we use TabSyn's original auto-encoder with default settings. Overall, we find that there is no strictly dominant encoding strategy that yields the best results (see Table 4).

Table 4: Results of encoding ablation for Gen-LRA on datasets and seeds from TabSyn. We find that there are is no strictly dominant encoding strategy for the attack.

| Encoding | AUC-ROC | TPR@FPR = 0.001 | TPR@FPR = 0.01 | TPR@FPR = 0.1 |
|---|---|---|---|---|
| Ordinal | **.583 (0.02)** | **0.040 (0.01)** | 0.06 (0.01) | 0.18 (0.04) |
| PCA | .557 (0.02) | 0.031 (0.01) | 0.042 (0.03) | **0.212(0.02)** |
| VAE | .577 (0.02) | 0.034 (0.01) | **0.052 (0.02)** | 0.209 (0.03) |

## D.2  ABLATION: DIFFERENT $k$ SIZES

Gen-LRA targets local overfitting by utilizing the $k$-nearest neighbors in $S$ to $x^*$. Consequently, $k$ serves as a hyperparameter in the attack. To assess the impact of $k$ on attack efficacy, we replicate the benchmarking experiments from Section 5 across varying values of $k$. The average AUC-ROC and corresponding standard deviations are reported in Table 5. Empirically, we observe that smaller values of $k$ generally enhance attack performance, though this effect varies by model. As discussed in Section 3, a global attack encompassing the entirety of $S$ is unlikely to yield significant membership signals. This is corroborated by the case where $k = N$, in which the AUC-ROC remains consistently at 0.5, underscoring that overfitting is inherently a localized phenomenon. These findings suggest that adversarial attacks on generative models should prioritize local regions to achieve effectiveness.

Table 5: Mean AUC-ROC at different $k$ values for Gen-LRA.

| Model | k=1 | k=3 | k=5 | k=10 | k=15 | k=20 | k=N |
|---|---|---|---|---|---|---|---|
| AdsGAN | 0.514 (0.02) | 0.518 (0.02) | 0.519 (0.02) | 0.520 (0.02) | 0.521 (0.02) | 0.521 (0.02) | 0.500 (0.00) |
| ARF | 0.532 (0.02) | 0.538 (0.02) | 0.540 (0.02) | 0.540 (0.03) | 0.540 (0.03) | 0.539 (0.03) | 0.500 (0.00) |
| Bayesian Network | 0.650 (0.07) | 0.645 (0.07) | 0.640 (0.07) | 0.634 (0.07) | 0.631 (0.07) | 0.629 (0.07) | 0.500 (0.00) |
| CTGAN | 0.514 (0.02) | 0.516 (0.02) | 0.517 (0.02) | 0.517 (0.02) | 0.518 (0.02) | 0.518 (0.02) | 0.500 (0.00) |
| Tab-DDPM | 0.595 (0.07) | 0.595 (0.07) | 0.594 (0.07) | 0.592 (0.06) | 0.591 (0.06) | 0.589 (0.06) | 0.500 (0.00 |
| Normalizing Flow | 0.503 (0.02) | 0.503 (0.02) | 0.505 (0.02) | 0.506 (0.02) | 0.506 (0.02) | 0.506 (0.02) | 0.500 (0.00) |
| TVAE | 0.527 (0.03) | 0.531 (0.03) | 0.531 (0.03) | 0.531 (0.03) | 0.530 (0.03) | 0.529 (0.03) | 0.500 (0.00) |

Table 6: Mean Accuracy for each Membership Inference Attack across model architectures and datasets.

| Model | Gen-LRA (Ours) | MC | DCR | DCR-Diff | DPI | DOMIAS | LOGAN 2017 |
|---|---|---|---|---|---|---|---|
| AdsGAN | **0.524 (0.02)** | 0.513 (0.02) | 0.513 (0.02) | 0.513 (0.02) | 0.515 (0.02) | 0.513 (0.02) | 0.503 (0.02) |
| ARF | **0.539 (0.02)** | 0.524 (0.02) | 0.524 (0.02) | 0.529 (0.02) | 0.526 (0.02) | 0.524 (0.02) | 0.503 (0.02) |
| Bayesian Network | 0.619 (0.05) | **0.629 (0.05)** | 0.629 (0.05) | 0.621 (0.05) | 0.538 (0.02) | 0.599 (0.05) | 0.504 (0.02) |
| CTGAN | **0.523 (0.02)** | 0.509 (0.02) | 0.509 (0.02) | 0.511 (0.02) | 0.513 (0.02) | 0.511 (0.02) | 0.504 (0.02) |
| Tab-DDPM | **0.58 (0.04)** | 0.564 (0.05) | 0.564 (0.05) | 0.563 (0.05) | 0.537 (0.02) | 0.563 (0.04) | 0.504 (0.02) |
| Normalizing Flows | **0.517 (0.02)** | 0.504 (0.02) | 0.504 (0.02) | 0.504 (0.02) | 0.505 (0.02) | 0.504 (0.02) | 0.501 (0.02) |
| PATEGAN | **0.514 (0.02)** | 0.501 (0.02) | 0.501 (0.02) | 0.499 (0.02) | 0.499 (0.02) | 0.500 (0.02) | 0.501 (0.02) |
| TVAE | **0.533 (0.02)** | 0.520 (0.02) | 0.520 (0.02) | 0.522 (0.02) | 0.517 (0.02) | 0.518 (0.02) | 0.503 (0.02) |
| **Rank** | **1.3** | 3.2 | 3.4 | 3.6 | 3.6 | 3.9 | 5.5 |

## D.3 THRESHOLDING/ ACCURACY REPORTING

We report the mean accuracy of the results of our main exeriment. Here, to create a comparable thresholding decision for each attack, we take the median of the scores across each test set. While we do not recommend in practice considering the accuracy of the attack as it is likely to under-represent privacy leakage, we still showcase that even with a simple threshold rule, Gen-LRA usually performs well.

