# OpenReview forum: "Privacy Auditing Synthetic Data Release through Local Likelihood Attacks"
_ICLR.cc/2026/Conference — ICLR 2026 Conference Withdrawn Submission_

### Official Review · Reviewer_bNoy · 2025-10-27

**Soundness:** 3
**Presentation:** 2
**Contribution:** 2
**Rating:** 4
**Confidence:** 4

**Summary:**

The proposes a new MIA methodology to audit the privacy risks associated with tabular synthetic data. Prior work has primarily focused on MIAs that are based on distance or density heuristics, or make additional assumptions for an attacker that are not realistic.

Specifically, the paper considers a 'no box' threat model, making no assumptions for an adversary when it comes to model access or knowledge about the model architecture or its training process. Instead, the adversary infers membership only informed by the released synthetic data and access to some auxiliary dataset. Authors propose Gen-LRA, a technique that compute, for each record, a score inspired by influence functions that estimate the influence that the target record has had on the generated synthetic data, following a surrogate model. As surrogate model, they find that Gaussian KDEs work the best and are computationally efficient.

They evaluate their method on 15 datasets and 9 generative models, comparing it against 7 other MIAs, and find it to be typically ranked as the best method across setups.  They evaluate this both using AUC and TPR at low values of FPR, consistent with the literature. Finally, they compare the performance of their method using KDE versus deep learning based estimators, finding that KDEs consistently work better.

**Strengths:**

- The paper studies an interesting problem, i.e. auditing privacy risk of synthetic tabular data.
- In contrast to prior work, this paper considers the more realistic no-box threat model.
- Authors consider a substantial evaluation setup, with 15 datasets and 9 synthetic data generators. Across setups, their method outperforms 7 baseline MIAs under the same threat model.

**Weaknesses:**

The paper might be overclaiming the success of their method. Namely, inspecting table 2, we find that for a substantial number of setups:

(1) the MIA AUC is very close to a random guess (<0.55), which might be too low to draw meaningful insights when it comes to the rank of a method.

(2) the MIA AUC for Gen-LRA only marginally outperforms the other MIAs, with in many cases the confidence intervals overlapping.

As such, the reported rank in Table 1 might be a misleading metric, and the results may be less significant as being claimed.

Further:
- While no-box, the threat model still assumes access to an auxiliary dataset (but justifiable).
- In section 4.1, when describing DCR, it would make sense to cite this recent piece of work [1].
- While the paper's method indeed outperforms all baselines under the same threat model, it would still be valuable to compare the performance of Gen-LRA to the shadow model based MIAs from for instance Stadler et al. (2022); Houssiau et al. (2022); Meeus et al. (2024). This would further ground the contribution in related work, and lead to a deeper understanding of the significance of the relaxations made in the threat model (i.e. no knowledge of the model training, and less computational resources).

[1] Yao, Z., Krčo, N., Ganev, G., & de Montjoye, Y. A. (2025, September). The DCR delusion: Measuring the privacy risk of synthetic data. In European Symposium on Research in Computer Security (pp. 469-487). Cham: Springer Nature Switzerland.

**Questions:**

- While MIA AUCs for Gen-LRA from Table 2 might not all be significantly higher than the ones for other methods, the TPR at low FPR values for table 3 seem to be more significant. Could authors provide the full ROC curves (in log-log scale) to better understand the results?
- How exactly do you evaluate the MIAs in Table 3? How many members, non-members do you use? I don't believe it's been made explicit in Section 5.1.
- How do you compute the k nearest neighbors, i.e. what kind of distance metric do you use?
- Are the utility metrics or findings from Figure 2 explained?
- Could authors add an analysis on the vulnerability of records? Currently the paper claims that Gen-LRA measures the degree of local overfitting, but it could be interesting to further analyze and show that this is the case, and when it happens. For instance, does this happen more for outliers, in line with Meeus et al.?

Minor:
- Line 50 (while 'a' promising)
- Line 412: 'to measures'

---

### Official Review · Reviewer_rE2s · 2025-10-30

**Soundness:** 2
**Presentation:** 2
**Contribution:** 2
**Rating:** 2
**Confidence:** 5

**Summary:**

The paper proposes a new Membership Inference Attack (MIA), Gen-LRA, designed for tabular synthetic data under a no-box threat model.
In this setting, the adversary has access only to the released synthetic dataset and a reference dataset drawn from the same distribution as the generative model’s training data.
The attack attempts to detect overfitting by estimating the effect of a target record through a local likelihood ratio -- comparing density estimates computed with and without the target record, using its nearest neighbors in the synthetic data.
Gen-LRA performs better compared to MIAs in the same no-box setting.

**Strengths:**

* Gen-LRA operates in a restricted no-box threat model, requiring access only to a single released synthetic dataset (and a reference dataset from the same distribution).
* Gen-LRA is conceptually sound, drawing on established ideas from prior privacy attacks such as DOMIAS (density estimation), DCR (distance to closest records), and LiRA (likelihood ratios with and without the target record).
* Gen-LRA is efficient/practical requiring only fitting of k-nearest neighbors and two kernel density estimators, without training several shadow models.
* Gen-LRA is benchmarked against a wide range of baseline MIAs (under the same no-box assumptions), datasets, and target generative models.
* Gen-LRA consistently outperforms baseline MIAs under the same no-box assumptions.

**Weaknesses:**

## Weaknesses:

* I struggle with the paper's framing of Gen-LRA as a "privacy auditing" method rather than as a MIA. In an auditing context, there is typically an explicit privacy claim (or bound) to be verified/challenged (i.e., a DP guarantee) against which empirical leakage can be measured.
For instance, DP auditing is an established area of research, where the goal is to empirically estimate the realized privacy loss and compare it to the theoretical bound. Prominent examples include studies in ML settings [1, 2, 3, 4] and in synthetic tabular data (Houssiau et al., 2022; [6]).
In contrast, Gen-LRA evaluates privacy leakage in synthetic data without an explicit privacy claim to audit.
As a result, it is not entirely clear what "auditing" means in this context.

* Connected to the previous point, previous MIA approaches such as (Stadler et al., 2022; Houssiau et al., 2022; Meeus et al., 2024) that rely on shadow modeling are described as having "unreasonable/unrealistic assumptions."
However, this framing is misleading -- such methods are typically evaluated within well-defined DP threat models, where access to the target model, shadow models, auxiliary data, etc. is perfectly valid.
Furthermore, the paper’s statement that Golob et al. (2024) show significant privacy leakage in DP synthetic data generation due to releasing the model architecture/implementation is also misleading.
Under a DP threat model, releasing the model implementation does not increase privacy leakage beyond the DP guarantee/bound.
In fact, DP inherently accounts for potential future information leakage/auxiliary knowledge that an adversary might gain (which is not captured under the no-box setting), and publishing DP mechanisms promotes transparency and helps ensure that implementations are free from hidden bugs/misconfigurations.

* While Gen-LRA outperforms baseline MIAs under the same no-box assumptions in terms of ranking, its relative performance does not seem statistically significantly different compared to DCR, which the paper describes as "exhibiting limited capability."
For instance, in Table 2, Gen-LRA and DCR have very high correlation and their scores are within half a standard deviation of each other (i.e., Gen-LRA - 0.5 sd < DCR + 0.5 sd).
Moreover, for TabSyn, Gen-LRA is highlighted as best attack (AUC = 0.583), even though DCR achieves a slightly higher AUC (0.585).
This raises concerns, given that the paper acknowledges DCR’s inability to capture privacy risk and prior work has shown that DCR scores do not correlate with state-of-the-art MIAs [7].

* Additionally, the absolute results of Gen-LRA are not discussed or interpreted. For example, in Table 2, excluding Bayesian Network, the average AUC of Gen-LRA is around 0.55.
It remains unclear whether this level of performance presents meaningful privacy leakage, or whether the evaluated synthetic dataset/generative models should be considered private.

* It is unclear how much performance is lost by operating in the no-box setting compared to black-box (ability to train shadow models).
While the paper argues that shadow modeling is impractical for large diffusion and language models, only two target models (Tab-DDPM and TabSyn) are diffusion-based.
As a result, comparison with state-of-the-art black-box MIAs is limited. For instance:
  * Stadler et al., 2022 detects privacy violations in PATE-GAN, whereas Gen-LRA reports PATE-GAN as the most private model (AUC = 0.520).
  * Meeus et al., 2024 achieves AUCs of 0.858/0.810 vs. Bayesian Network, compared to Gen-LRA’s 0.679.
  * ([5] removes the assumption for access to auxiliary data and still achieves accuracies of 0.618/0.644 vs. Bayesian Network.)
  * [7] achieves AUC of 0.94 vs. most vulnerable target records in CTGAN, compared to Gen-LRA’s 0.533.

## Minor Weaknesses/Comments:
* The paper contains a number of formatting errors that reduce clarity:
  * References are inconsistently cited (e.g., \citet vs. \cite/\citep).
  * Appendices are referred to as "Appendix 1, 2, 3, 4," but the actual appendices in the paper are labeled A, B, C, D, and hyperlinks do not work.
  * The description of TabSyn spans two separate sections (Appendix C.2 and the first paragraph of C.3).
* The naming of Gen-LRA appears heavily influenced by LiRA (Carlini et al., 2021), which may be confusing since the original attack is called LiRA, not LRA.
* Records under attack are typically referred to as "target records," not "test observations."
* There are several typos -- "while a promising," "ususally," "adhoc" vs. "ad-hoc," "overfitness," etc.
* The code is not publicly available; the linked repository is empty.

##  References:

[1] Jagielski et al., Private Machine Learning: How Private is Private SGD? In NeurIPS, 2020.

[2] Nasr et al., Adversary Instantiation: Lower Bounds for Differentially Private Machine Learning. In IEEE S&P, 2021.

[3] Nasr et al., Tight Auditing of Differentially Private Machine Learning. In USENIX Security, 2023.

[4] Steinke et al., Privacy Auditing with One (1) Training Run. In NeurIPS, 2023.

[5] Guépin et al., Synthetic Is All You Need: Removing the Auxiliary Data Assumption for Membership Inference Attacks Against Synthetic Data. In DCM at ESORICS, 2023

[6] Annamalai et al., "What do you want from theory alone?" Experimenting with Tight Auditing of Differentially Private Synthetic Data Generation. In USENIX Security, 2024

[7] Yao et al., The DCR Delusion: Measuring the Privacy Risk of Synthetic Data. In ESORICS, 2025

**Questions:**

Please refer to weaknesses, additionally:
* Could the authors clarify what "privacy auditing" means in the context of Gen-LRA?
* Could the authors discuss the high correlation and close performance between Gen-LRA and DCR, and what this implies for the Gen-LRA's practical utility?
* Could the authors discuss/interpret the absolute performance of Gen-LRA (e.g., AUC around 0.55 for most models)? Does this level of performance indicate meaningful privacy leakage?
* Could the authors discuss the performance gap between Gen-LRA (no-box) and state-of-the-art MIAs under black-box threat models?
* Could the authors comment on potential mitigation strategies for Gen-LRA?
* The authors note that Average Negative Log Likelihood and Negative Evidence Lower Bound often outperform KDE/BNAF -- is this also the case for Gen-LRA?
* Could the authors clarify the definition of the holdout set H in Section 5.1? Is this equivalent to X_test in Algorithm 1?

---

### Official Review · Reviewer_6SCL · 2025-10-30

**Soundness:** 2
**Presentation:** 2
**Contribution:** 2
**Rating:** 2
**Confidence:** 4

**Summary:**

This paper proposes Gen-LRA, a model-agnostic Membership Inference Attack against tabular synthetic data generators. Specifically, their attack compares the local density around a target point x* of the synthetic dataset S, to that of a reference dataset R and also R U x*. The authors claim that if S is overfitted on the target point, the local density of S around x* should be more similar to R U x* than R. They evaluate their attack against six other MIAs across nine synthetic data generator architectures and fifteen datasets.

**Strengths:**

- The paper creatively combines different existing techniques to produce a new MIA against tabular synthetic data; namely construction of shadow datasets and targeting overfitting using density estimation.
- The paper explores multiple density estimation techniques in addition to the SOTA and provides useful insight on the tradeoffs and performance of different techniques for overfitting detection.
- The paper provides a novel encoding scheme for KDE for usage in density estimation of tabular data.

**Weaknesses:**

Please refer to the questions section for further elaboration on the following points.
- The paper appears to overclaim the improvement offered by Gen-LRA. Gen-LRA assumes a weak attacker and only shows marginal improvement over a limited list of MIAs. I am also not convinced that an MIA obtaining AUC values so close to 0.5 can be considered successful.
- The paper requires additional justification for their assumption that privacy leakage can be evaluated exclusively through local overfitting detection.
- The paper includes diffusion models TabDDPM and TabSyn in their analysis, but does not include diffusion-specific attacks in their comparisons.
- The paper is missing several important citations
Minor:
- References to the Appendix do not match the Appendix section titles (e.g., Appendix D.1 is referred to as Appendix 4.1)

**Questions:**

Overclaim
- The paper claims that Gen-LRA “outperforms” competing MIA methods and “excels”, however Table 2 and 3 show only marginal improvements. Furthermore, it has been shown by [3] and [6] that DCR is a poor measure of privacy leakage in synthetic data, and Gen-LRA only appears to offer marginal improvements over DCR. Could you substantiate your claim on Gen-LRA’s performance further, and evaluate the MIAs’ performance in more detail, specifically when it comes to performance on outliers?
- Table 1’s usage of ranking as a metric inflates the apparent gains that Gen-LRA obtains. Table 1 also directly contradicts Table 2, as Table 1 indicates that all other MIA methods perform similarly, while Table 2 shows that LOGAN performs little better than random guessing on all synthetic data generators. Can you explain why you believe the relative rank of a method is more informative about the method’s performance than TPR at a fixed FPR or AUC? Can you also explain the described discrepancy in your results?
- The paper assumes a weak attacker with no knowledge about the target model, which differs from the black-box assumption of SOTA MIAs on tabular synthetic data. It is not good practice to assume that the attacker is unable to gain any information about the target model architecture, especially given the existence of model extraction attacks on GANs [1] and diffusion models [2]. Furthermore, it has been shown by [8] that even mismatched architecture between the shadow and target models can lead to a successful MIA. Could you justify more clearly the choice for such an assumption?
- Also, can you show the MIA performance tradeoff of such an assumption by comparing Gen-LRA to the state-of-the-art MIA for tabular synthetic data (TAPAS extended for continuous data [7]) experimentally?
- The paper claims that Gen-LRA is practical from a computational cost perspective, however no experimental data is provided. How does Gen-LRA compare computational cost wise to other MIAs (including extended TAPAS)?
- The paper also averages out the performance of each MIA across all datasets, which each have different data distributions. Averaging the performance across datasets can mask the performance of MIAs on more vulnerable datasets. Could you report the performance of all MIAs on each dataset respectively?
- Lastly, in section 3.3, the paper mentions that KDE was selected as it achieves better performance than other surrogate models such as tractable probabilistic models and Baynessian methods. Could you provide experimental data substantiating this claim. Also, can you substantiate your choice of threshold in section 3.3 with ablation studies to show how the performance of Gen-LRA varies with different thresholds?

Localization Assumption
- The paper focuses exclusively on local overfitting detection to assess privacy risk. However, recent work in [3], shows that privacy leakage can occur outside of the locality of the target point. How would Gen-LRA be able to detect such instances of privacy leakage, such as the case with CTGAN in [3]?
- I would like to see experiments comparing KDE to BNAE specifically on IndHist. IndHist is not a SOTA synthetic generator, but it would provide useful insight on the localization assumption, as it is a synthetic generator that does not preserve the joint probability distribution between features.

Lack of Diffusion Specific MIAs
- [5] introduce attacks (including one where the attacker only has access to published synthetic data of the target model) that perform well. Including these as baselines would give context and help to interpret the performance of Gen-LRA for diffusion models.
Missing Citations
- The paper fails to cite [4], the first paper to present a systematic evaluation of MIAs on synthetic tabular data.
- The paper also fails to cite the source for the formal definition of the Membership Inference Attack Game.
- The paper claims that MIA metrics can obfuscate true performance of MIAs in Table 1, but provides no citation for this claim.
[1] Hu, Hailong, et al. “Model Extraction and Defenses on Generative Adversarial Networks.” ACSAC 2021
[2] Carlini, Nicholas, et al. “Extracting Training Data from Diffusion Models.” USENIX Security 2023
[3] Yao, Zexi, et al. "The DCR Delusion: Measuring the Privacy Risk of Synthetic Data." ESORICS 2025
[4] Stadler, Theresa, et al. “Synthetic Data – Anonymisation Groundhog Day.” USENIX Security 2022
[5] Wu, Xiaoyu, et al. "Winning the midst challenge: New membership inference attacks on diffusion models for tabular data synthesis." arXiv preprint arXiv:2503.12008 (2025).
[6] Ganev, Georgi, et al. “The Inadequacy of Similarity-based Privacy Metrics: Privacy Attacks against "Truly Anonymous" Synthetic Datasets.” IEEE S&P 2025
[7] Meeus, Matthieu, et al. “Achilles' Heels: Vulnerable Record Identification in Synthetic Data Publishing.” ESORICS 2023
[8] Carlini, Nicholas, et al. “Membership Inference Attacks From First Principles” IEEE S&P 2022

---

### Official Review · Reviewer_gTeM · 2025-11-01

**Soundness:** 3
**Presentation:** 4
**Contribution:** 3
**Rating:** 6
**Confidence:** 3

**Summary:**

This paper introduces Gen-LRA, a no-box membership inference attack (MIA) for tabular synthetic data. The central idea is to measure the influence of a candidate point on a surrogate estimate of the likelihood of the released synthetic data $S$. The algorithm is localized by considering the $k$-nearest elements in $S$ to the data point, and instantiated with Gaussian Kernel Density Estimators (KDE) as surrogate models. Across a broad benchmark (15 datasets and 9 generators), Gen-LRA generally outperforms prior MIAs.

**Strengths:**

1. The method is very simple, elegant and easy to implement. The idea of considering the influence of adding a data point to target overfitting is natural.
2. The method is supported by strong empirical evidence, showing that it outperforms other SOTA methods under multiple settings, especially in the low FPR regime.

**Weaknesses:**

1. The MIA game assumes access to a reference set $R$, roughly as large as the training set $T$. This hypothesis might not be realistic in practical scenarios. In practice, auditors may have access to smaller datasets that might not have the same distribution as the training set.
2. The datasets $R,T$ are drawn from some distribution $\mu$. This setting might hold for practical auditing of synthetic data generators in some realistic scenarios. However, this method cannot be used to audit the privacy guarantees of synthetic data generators. In fact, such procedures typically require selecting worst case datasets and test points.

**Questions:**

1. It would be nice to check the performance of Gen-LRA under small distribution shifts of $R$.
2. In 3.2, it is stated that "the likelihood ratio is invariant of the encoding of the data". However, Theorem 3.1 only provides this guarantee for invertible encodings. In Appendix D.1, you perform PCA, which is not invertible. Can you clarify on this?

---

### Note · Authors · 2025-11-30

I have read and agree with the venue's withdrawal policy on behalf of myself and my co-authors.